# Computational Fluid Dynamics Simulation of Suspended Solids Transport in a Secondary Facultative Lagoon Used for Wastewater Treatment

Andres Mauricio Zapata Rivera [1,*] , Joel Ducoste [2] , Miguel Ricardo Peña [3] and Margarita Portapila [4]

1 Departamento de Energía, Universidad de la Costa CUC, Calle 58 # 55-66, Barranquilla 080002, Colombia
2 Department of Civil, Construction and Environmental Engineering, North Carolina State University, 208 Mann Hall, Campus Box 7908, Raleigh, NC 27696-7908, USA; jducoste@ncsu.edu
3 Instituto Cinara, Universidad del Valle, Calle 13 # 100-00 Edificio 341, Cali 76001, Colombia; miguel.pena@correounivalle.edu.co
4 Centro Internacional Franco Argentino de Ciencias de la Información y de Sistemas CIFASIS-CONICET-UNR, Ocampo y Esmeralda S2000 EZP, Rosario S2000EZP, Argentina; margarita.portapila@gmail.com
* Correspondence: andres.zapata@correounivalle.edu.co; Tel.: +57-315-5174098

**Abstract:** The facultative lagoon hydrodynamics has been evaluated using computational fluid dynamics tools, however, little progress has been made in describing the transport of suspended solids within these systems, and their effects on fluid hydrodynamics. Traditionally, CFD models have been built using pure water. In this sense, the novelty in this study was to evaluate the influence of suspended solids transport on the hydrodynamics of an facultative lagoon. Two three-dimensional CFD models were developed, a single-phase model (pure water) and a two-phase model (water and suspended solids), for a conventional FL in Ginebra, Valle del Cauca, Colombia. Model results were compared with experimental tracer studies, displaying different tracer dispersion characteristics. Differences in the fluid velocity field were identified when suspended solids were added to the simulation. The fluid velocities in the single-phase model were greater than the fluid velocities obtained in the two-phase model, (0.127 m·s$^{-1}$ and 0.115 m·s$^{-1}$, respectively). Additionally, the dispersion number of each model showed that the single-phase model (0.478) exhibited a better behavior of complete mixing reactor than the two-phase model (0.403). These results can be attributed to the effect of the drag and slip forces of the solids on the velocity of the fluid. In conclusion, the fluid of FL in these models is better represented as a two-phase fluid in which the particle–fluid interactions are represented by drag and slip forces.

**Keywords:** CFD; hydrodynamics; single-phase model; suspended solids transport; tracer test; two-phase model

## 1. Introduction

Facultative lagoons (FLs) are among the most widely used technologies for wastewater (WW) remediation globally [1]. They are inexpensive, efficient, sustainable, simple to design, easy to operate and easy to maintain [2–4]. The design characteristics of these systems favor the development of a wide variety of microorganisms and processes, such as nitrification, ammonification, denitrification, phosphorus removal by assimilation into biomass and precipitation, methanogenesis and photosynthesis [5–7]. All these processes can work collectively towards the treatment of complex contaminated waters in FLs [8].

The hydrodynamics of FLs used in the treatment of WW are critical to their design, and significantly impact the performance of these constructed ecosystems [9]. Therefore, the operation and design of an FL require an understanding of its hydrodynamics behavior [10], which is complex and can include fluid recirculation patterns, dead zones, physical structures such as baffles or screens, and, in some cases, mechanical aeration devices [11–13]. In addition, the hydrodynamics of an FL can be influenced by the presence

of suspended solids and the population of algae generated in the lagoon. In this sense, a limited number of research studies have been performed that evaluated the impact of suspended solids on the hydrodynamics of FLs.

Alvarado et al. built a CFD model to study the relation between velocity profiles and sludge deposition during 10 years of operation of the Ucubamba WSP in Cuenca-Ecuador. Three sludge accumulation scenarios based on bathymetric surveys were simulated. For this, three different geometries and meshes were created, corresponding to each sludge accumulation scenario. The pond was modeled as a tank and the presence of sludge was accounted for by removing the corresponding volume of the pond where sludge accumulation had been measured as modifying the geometry and mesh. Alvarado et al., found that sludge accumulation patterns and velocity profiles are interrelated and directly affect pond hydraulic performance.

Ouedrago et al. simulated the flow and tracer transport using a numerical solver of the three-dimensional Reynolds-averaged Navier–Stokes equations (RANS). The sludge geometry, as well as pond geometry and water flow parameters obtained in the field were used to model the pond. The RANS solver was used to predict the hydraulic performance of the WSP under future sludge accumulation scenarios. Ouedrago et al. demonstrated that an increase in sludge volume (depending on the sludge distribution or geometry) may improve the hydraulic performance of a WSP, by inducing a baffling effect. In both cases, the CFD models that were reported used the profile of settled solids as a function of the lagoon geometry, but did not characterize the transport of suspended solids within the lagoon and their impact on the fluid hydrodynamics and velocity profiles [14,15]. One reason for this lack of research on the interaction between solid transport and FL fluid mechanics may be the limited qualitative and quantitative techniques available to characterize these interactions [16]. The existing techniques that can be applied to lagoon systems used for WW treatment are either through the use of experimental tracer studies or more recently the use of computational fluid dynamics (CFD) techniques.

Tracer studies are well described in the literature, and are one of the most widely used methods to evaluate the mixing characteristics in lagoons, despite the demanding fieldwork required to use them. However, tracer studies provide only a "black-box" evaluation of the fluid mechanics, where there is limited inference about the fluid flow patterns inside the lagoon [17]. CFD models that employ finite element or finite volume numerical methods, on the other hand, are powerful and innovative tools to study the details of the fluid mechanics in lagoon systems used for WW treatment [18]. Studies involving the use of CFD in characterizing the fluid mechanics' behavior of lagoons have been limited to single-phase analyses [13,19], or to explain the solids deposition profile at the bottom of the lagoon [14,15]. The effects of deposited solids on the fluid hydraulics and the velocity field have been analyzed by modifying the computational domain of the models [15]. While both research studies used CFD to either explain the solids deposition pattern or the flow pattern when a certain solids deposition is assumed, research is needed to understand the influences of solids transport on the overall mixing and fluid flow pattern in FLs.

In this study, two three-dimensional (3D) CFD models were developed for an FL. Two scenarios were compared: (1) a single-phase model that only simulates pure water as the fluid, and (2) a two-phase model that considers a mixture of water and suspended solids. The novelty of the two-phase CFD model consists of using methods, such as the mixing model and the drag and slip force model to describe the dispersion of suspended solids within the facultative lagoon. The model was validated with experimental tracer tests using rhodamine WT (RWT) and values of suspended solids concentrations. This research contributes to the study of the phenomena of solids transport in an FL, and its effects on fluid hydrodynamics, as well as its potential impact on the FL process performance. The two-phase scenario can be used to predict and identify the location in which particles are likely to settle, and the settling rates of those particles. Furthermore, the two-phase CFD model can be used to simulate the transport of organic pollutants, such as flame retardants or polychlorinated biphenyls because they are effective to transport these kinds

of hydrophobic pollutants [20,21]. Finally, it can be used for the evaluation of possible design modifications and their potential impact on solids settling in the FL.

## 2. Material and Methods

### 2.1. Location of the Conventional Facultative Lagoon

Experimental research was performed in a conventional secondary facultative lagoon, located in the Wastewater and Recycling Research Station in the municipality of Ginebra-Valle del Cauca, Colombia at $3°43'50''$ N and $76°16'20''$ E, at 1040 m above sea level. The average temperature in this municipality is 23 °C and the average annual rainfall is 1280 mm, corresponding to group A (tropical) according to the Köppen classification [22]. A schematic of the experimental system is shown in [21]. The inflow of wastewater into the FL was 23.76 $m^3 \cdot d^{-1}$ and was regulated using an automatic globe valve (KSB SE & CO, Frankenthal, Germany). The inlet and outlet structures were submerged tubes 0.0546 m in diameter and were oriented parallel to the longest side of the lagoon. The surface area of this constructed ecosystem is 83.22 $m^2$, and has the following dimensions: depth = 1.48 m, width = 5.70 m and length = 14.60 m. The operating flow rate produces a hydraulic retention time of 3.99 days. The FL was designed for an organic load of 279 kg BOD$\cdot ha^{-1} \cdot d^{-1}$.

### 2.2. Experimental Tracer Studies

A tracer study was performed to validate the mixing characteristics of the FL. Rhodamine WT (RWT) was used as a tracer, using the pulse input technique. For this purpose, 3.8 g of tracer was added to a 1000 mL volumetric flask with wastewater from the FL, to ensure the tracer solution maintained the same temperature of the wastewater [17]. The resulting solution was quickly added to the FL inlet. The tracer concentration was measured "on-site" in the effluent using a fluorescence detection technique. For this purpose, a Turner M 8000-010 fluorometer (Turner Designs Company, Sunnyvale, CA, USA) was used with a linear detection range between 0.4 $\mu g \cdot kg^{-1}$ and 300 $\mu g \cdot kg^{-1}$ and wavelengths between 550–570 nm. The data were used to reconstruct the experimental residence time density function for the FL. In the CFD model, the tracer injection was implemented using the chemical species model (Species) and the pulse input method. The output concentration was monitored using an "area-weighted averaged concentration" monitor, taking into account the velocity distribution across the cross-section [23].

### 2.3. Experimental Suspended Solid Concentrations

The suspended solid concentration profiles for the model validation were determined as suggested by [24]. For this purpose, three measurement campaigns were carried out. The outflowing SS concentrations and three points within the FL located at point P2 (L/2) and its respective depths 0.05 m, 0.45 m and 1.40 m were continuously measured using an optical turbidimeter (TU5300sc Hach® Loveland, CO, USA).

### 2.4. Validation of Data from Research

The normality of the simulated and experimental data series was validated by applying the Shapiro–Wilks test ($n > 50$), where a $p > 0.05$ indicates normality. Additionally, the variances of the residence time distribution curves (RTD) were compared using the Levene test ($n > 50$). In this case, a $p < 0.05$ indicates that there are no statistically significant differences between the variances. Accuracy tests were performed to compare the simulated suspended solids concentrations and the experimental data. These accuracy tests include the calculation of the absolute mean deviation (AMD%) and the relative mean deviation (RMD%) as measures of the error between the simulated and experimental SS concentrations; and the sum of the squared errors (SSE), the root mean square error (RMSE) and root mean square deviation (RMSD) used as an accuracy test for the SS concentration data series. The validation followed the recommendations of [25,26]

## 3. CFD Model Specifications

### 3.1. Geometry and Discretization

The 3D CFD models were developed using the software ANSYS Fluent®, Release 16.1 on a Dell Precision TX3500 workstation with an Intel®Xeon® X3470 processor (8 MB Cache, 2.93 GHz, Turbo, HT). The 3D geometry of the experimental FL was built using ANSYS Design Modeler® software Release 16.1. The finite volume method was used for discretization. The computational domain was divided into 161,890 hexagonal elements of 0.05 m, using the Ansys Inc® ICEM CFD™ meshing software Release 16.1. The quality of the mesh elements was evaluated using the determinant and internal angle methods. The first method guarantees an element quality of greater than 0.5, and the second an internal angle greater than 9°. This procedure was performed to favor solution accuracy and model convergence. Figure 1 displays the geometry and mesh of the CFD model.

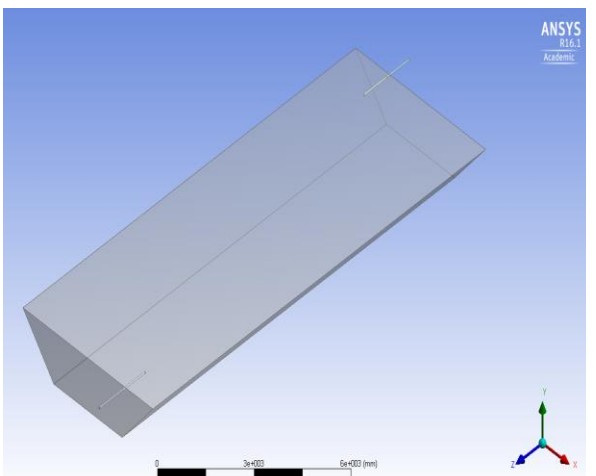 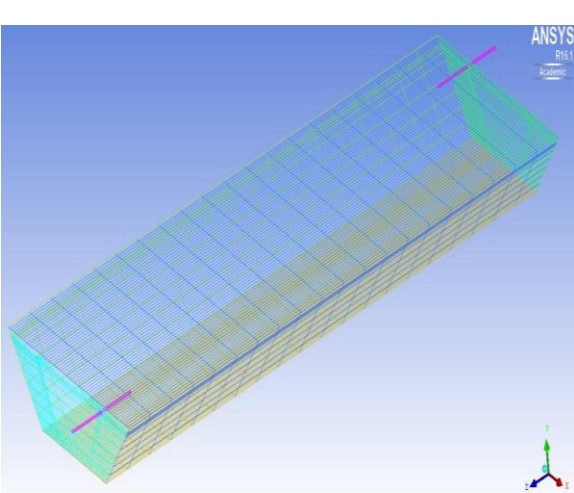

**Figure 1.** Geometry and mesh used in CFD model.

### 3.2. Boundary Conditions

Boundary conditions used in the CFD model are shown in Table 1. At the inlet, the boundary condition "velocity inlet" establishes velocity vectors and scalar properties of the fluid in the FL influent. The "outflow" boundary, with a mass imbalance of 0%, was applied to the single-phase model. Subsequently, when coupling the Eulerian model for the two-phase model, the outlet condition was changed to "pressure outlet" because the coupling of the two-phase model is not compatible with the boundary condition "outflow" [23]. The boundary "wall" was applied to the walls, to simulate the solid boundary conditions for viscous fluids. Two concentrations of suspended solids for the two-phase model were used. The average of the experimental data measured over 2 years and the probability distribution of this data series were calculated using a user's defined function (UDF). The probability distribution has a rectangular distribution with minimum and maximum values of 0.00181 and 0.0783% *w/v*. The concentration of suspended solids did not include the algae biomass generated in the interior of the lagoon.

**Table 1.** Boundary conditions used in CFD model.

| Zone | Boundary | Value | Units | Observations |
|---|---|---|---|---|
| Inlet | Velocity inlet | 0.115 | m·s$^{-1}$ | Turbulence Intensity = 5.29% <br> Dh = 0.0546 m <br> Re = 5822 <br> I = 0.053% <br> k = 5.78 × 10$^{-5}$ J·kg$^{-1}$ <br> $\varepsilon$ = 8.00 × 10$^{-6}$ m$^2$·s$^{-3}$ <br> Conc.: 0.041% *w/v*, UDF |
| Outlet | Outflow | 1.0 Fraction | N.A | Turbulence Intensity = 7.3% <br> Re = 506 <br> I = 0.073 (%) <br> k = 8.00 × 10$^{-7}$ J·kg$^{-1}$ <br> $\varepsilon$ = 1.00 × 10$^{-8}$ m$^2$·s$^{-3}$ |
| Walls | Stationary Wall | — | N.A | Polyethylene of low density |
| Surface | Free surface | 0.81 | m·s$^{-1}$ | The prevailing direction of the wind was used with an average speed of 0.81 m·s$^{-1}$ and northeast (NE) direction. |

Dh: hydraulic diameter; Re: Reynolds number; I: turbulence intensity; k and $\varepsilon$: parameters of the turbulence model; Conc.: suspended solids concentration (weight/volume percent).

Walls in the CFD model were treated as stationary, and the shear condition selected for walls was the "no-slip". The FL air/water surface was selected as a "free surface-slip wall" relative to the adjacent cell zone, which was the fluid.

The user defined function in the inlet zone represented the transitory suspended solid concentration measured during two years. It was used to replace the scalar concentration of 0.041% *w/v*. The UDF built corresponded to a "Define Profile" as suggested by [23], (see Supplementary Materials).

### 3.3. Drag and Slip Forces Models

In the two-phase model, two approximations were used to simulate the solids: the "Eulerian" and "Mixture" models. Results showed that the "Mixture" model reduced the computational time by 13% (from 360 h to 313.2 h) and memory requirements by 8% (from 65.3. MB to 60.08 MB); therefore, this model was used for the simulations. The volume of fluid (VOF) model was not appropriate for the suspended solids transport simulations because the Courant number was greater than 250, presenting a solution divergent for the model. The drag and slip forces were modeled for the interactions between the phases. Five combinations of the different drag and slip force models were compared to assess their performance: (a) Wen Yu-Legendre Magnaudet, (b) Huilin Gidaspow-Legendre Magnaudet, (c) Gidaspow-Legendre Magnaudet, (d) Syamlal Obrien-Legendre Magnaudet and (e) Gibilaru-Legendre Magnaudet [23]. The Gibilaru-Legendre Magnaudet relationship was selected, since none of the models displayed significant differences in the predicted values of solids profile at the FL outlet (please see Supplementary Materials).

### 3.4. Properties of the Materials

The physical properties of the wastewater and the discrete phase (suspended solids) were determined. The WW was assumed to be incompressible, exhibiting Newtonian behavior. The WW density ($\rho$) was 1020 kg·m$^{-3}$ and the dynamic viscosity ($\nu$) was 0.0011 kg·m$^{-1}$·s$^{-1}$ [27]. The suspended solids density ($\rho$) was 1170 kg·m$^{-3}$ [28] and three-particle diameters ($1 \times 10^{-5}$, $5 \times 10^{-5}$ and $8 \times 10^{-5}$ m) were used for SS [29], which spans the potential range that would occur in the full-scale FL.

### 3.5. Governing Equations

The governing equations were based on the Navier–Stokes equation, solved under transient conditions for the fluid and the dispersed phase. The three-dimensional continuity,

momentum and turbulence equations are shown below (Equations (1)–(4)). The model and its equations were addressed using a Eulerian multiphase approach because it offered acceptable results compared with the experimental data and lower computational resources in the Lagrangian approach.

Continuity equation

$$\frac{\partial}{\partial t}\left(\alpha_q \delta_q\right) + \nabla\left(\alpha_q \delta_q \overrightarrow{V}_q\right) = \sum_{p=1}^{n}\left(m_{pq} - m_{qp}\right) + S_q \tag{1}$$

Equation for momentum

$$\frac{\partial}{\partial t} = \left(\rho \overrightarrow{v}\right) + \nabla.\left(\rho \overrightarrow{v}\overrightarrow{v}\right) = -\nabla p + \nabla.\left(\overline{\overline{\tau}}\right) + \rho \overrightarrow{g} + \overrightarrow{F} \tag{2}$$

Turbulence model

The $k - \varepsilon$ realizable turbulence model equations are as follows:

$$\frac{\partial}{\partial t}(\rho k) + \frac{\partial}{\partial x_j}(\rho k u_j) = \frac{\partial}{\partial x_j}\left[\left(\mu + \frac{\mu_t}{\sigma_k}\right)\frac{\partial k}{\partial x_j}\right] + G_k + G_b - \rho\varepsilon - Y_M + S_k \tag{3}$$

and,

$$\frac{\partial}{\partial t}(\rho\varepsilon) + \frac{\partial}{\partial x_j}(\rho\varepsilon u_j) = \frac{\partial}{\partial x_j}\left[\left(\mu + \frac{\mu_t}{\sigma_\varepsilon}\right)\frac{\partial\varepsilon}{\partial x_j}\right] + \rho C_1 S\varepsilon - \rho C_2\frac{\varepsilon^2}{k + \sqrt{v\varepsilon}} + C_{1\varepsilon}\frac{\varepsilon}{k}C_{3\varepsilon}G_b + S_\varepsilon \tag{4}$$

The following default turbulence coefficients in the model were used for this application: $C_1 = 1.44$, $C_2 = 1.92$ and $C_3 = 0.09$. More details on the formulated equations can be found in the literature [23,30]

### 3.6. Mesh Independence Test

A mesh independent solution for the CFD model was analyzed as part of this study. Three grid densities with different cell sizes were used in the CFD model. These grid densities included: (1) 16,160 elements producing an element size of 0.5 m, (2) 161,890 elements producing an element size of 0.05 m and (3) 276,470 elements producing an element size of 0.025 m. Velocity and suspended solids concentration distributions were used to evaluate the grid independence of the model predictions.

## 4. Results and Discussion

### 4.1. Mesh Independence Test

The results of each grid independence evaluation are shown in Table 2. The suspended solids concentrations predicted by mesh densities two and three are not significantly different. Based on this information and the mesh quality criteria applied (the determinant and the internal angle methods), grid density two was chosen as the optimum mesh for all simulations.

**Table 2.** Predicted effluent suspended solids concentration and velocity from each mesh evaluated in CFD model.

| Mesh No. | Mesh Cell Size (m) | Velocity of Fluid (m·s$^{-1}$) | Suspended Solids Concentration (% $w/v$) |
|:---:|:---:|:---:|:---:|
| 1 | 0.500 | 0.01340 | 0.2930 |
| 2 | 0.050 | 0.01120 | 0.2240 |
| 3 | 0.025 | 0.01123 | 0.2221 |

*4.2. Tracer Studies: CFD Models vs. Experimental Results*

The results of the tracer tests from the CFD models were compared and validated with the experimental tracer test results following the recommendations of [11]. Figure 2 compares the experimental mean residence time distribution (RTD) with the single-phase and the two-phase CFD model RTDs. The results show that a larger peak appears in the single-phase CFD model in the first 1.5 h, with a difference from the time obtained in the experimental tracer studies of 0.5 h. The peak for the two-phase model occurs at the same magnitude and time as in the field studies.

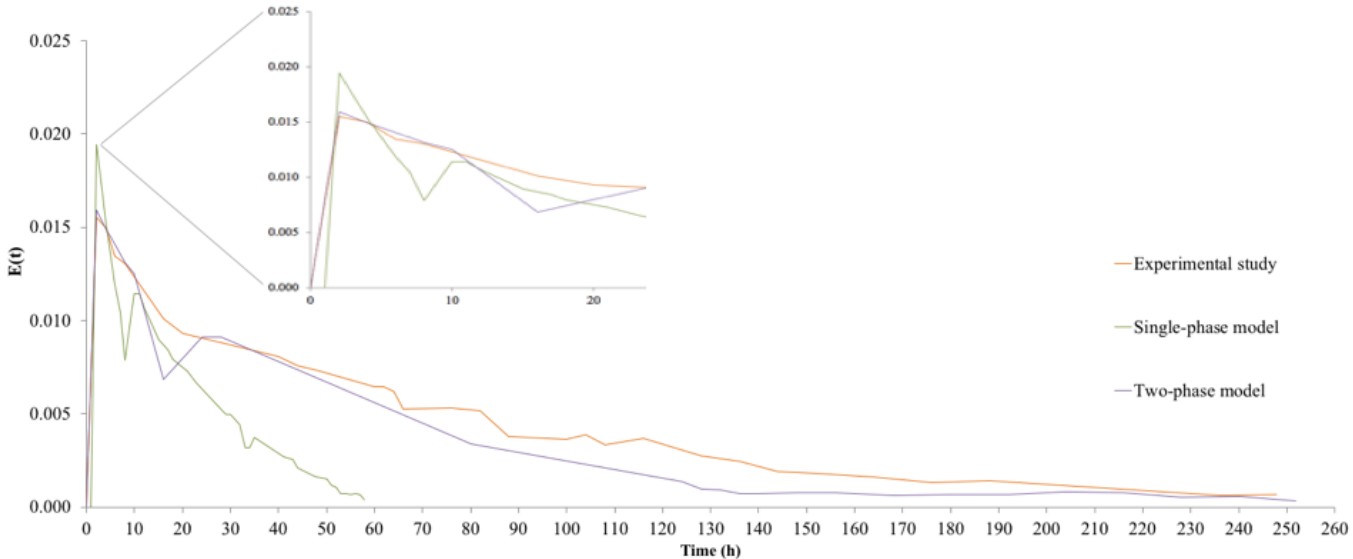

**Figure 2.** Residence time distribution obtained experimentally and with the CFD models (single-phase and two-phase).

The RTD curve from the single-phase model displays a shift to the lower time values and the lowest variance value ($\sigma^2 = 3057$) than the variances of the experimental data and two-phase model (see Table 3). The absence of suspended particles in the single-phase model changes the fluid velocity distribution in the FL. When solids are introduced into the flow domain, the interaction between the particles in suspension and the fluid (drag and slide forces) reduce the velocity distribution, and lead to greater mixing and dispersion. These results support the conclusions of [31] that showed multiphase CFD models that include appropriate particle–particle interactions and drag models significantly improve the final predictions of the solids transport in reactors.

**Table 3.** Parameters obtained from experimental tracer study and CFD models.

| Parameter | Study 1 | Single-Phase CFD Model | Two-Phase CFD Model |
|---|---|---|---|
| Experimental retention time (h) | 75 | 62 | 70 |
| Theoretical retention time (h) | 95.76 | 95.76 | 95.76 |
| Error (%) | 22 | 35 | 26 |
| Variance ($\sigma^2$) | 4602 | 3057 | 4607 |
| Dispersion number ($\delta$) | 0.436 | 0.478 | 0.403 |

The Shapiro–Wilk test ($n > 50$) showed that experimental and simulated data series did not correspond to a normal distribution ($p < 0.05$). The non-parametric Levene test was then applied to compare the variances of the experimental ($\sigma^2 = 4602$), single-phase ($\sigma^2 = 3057$) and two-phase model ($\sigma^2 = 4607$). Results of this analysis showed that there

was no statistically significant difference between the experimental and two-phase model variances ($p < 0.05$).

When the theoretical retention time was compared to the retention time obtained with the single-phase and two-phase models (see Table 2), an error of 35% was found in the single-phase model while an error of 26% was found in the two-phase model. Note that a 22% error was reported with the experimental data. Overall, both the experimental and simulated tracer tests demonstrate significant mixing occurring in this FL, and that there may be regions of unused or dead volume. Moreover, the simulated tracer results further indicate that a single-phase model overpredicts the mixing in the FL and that two-phase simulations should be used to characterize the mixing and hydraulic characteristics of FLs, so that proper design configurations can be explored with these models. The results of the accuracy tests showed that errors obtained for AMD% and RMD% between the two-phase model and the experimental data corresponded to 5% and 12%, respectively. The SSE was 0.730 and 0.870, respectively. The RMSE presented error percentages of 0.365 for the experimental data and 0.063 for the CFD model. Finally, the RMSD showed error percentages of 0.170 and 0.072 for the experimental and simulated data, indicating that the model presents less variability and greater robustness in the results.

### 4.3. Single-Phase CFD Model vs. Two-Phase CFD Model

Previous studies that have simulated the hydrodynamics behavior of this type of constructed ecosystem have recommended the standard k-ε model as one of the best descriptors of fluid hydrodynamics [14,15,32]. To select the model for this study, fluid behavior analysis was performed to determine if the fluid exhibited laminar, transitory or turbulent characteristics [23]. This aspect is not mentioned in the modeling articles reviewed; however, it is fundamental to the selection of the appropriate model. In this study, the Re was 5822 at the inlet, 506 at the outlet and 840 in the interior of the lagoon (transverse section). These data confirm that the selection of the realizable k-ε model was the best choice, since it is recommended for fluids in laminar or transitory regimes [23]. For the interactions between liquid and suspended solids, five combinations of the drag and slip forces models were tested, and the results showed the same solutions for the velocity fields, number of iterations and solution convergence. Hence, the "Gibilaru-Legendre, Magnaudet" combination was selected. Simulations with three particle diameters for the suspended solids were performed, and numerical results showed a difference of 3.7% in the suspended solids concentration in the outflow between the small and large diameters.

The CFD models describe the presence of short circuits and dead zones in the FL, which influence the quality of the effluent [11]. The streamlines for both scenarios (single-phase and two-phase) are shown in Figure 2. Two fluid re-circulation regions (zone 1, x: 1.35 m, z: 13.5 m; zone 2, x: 2.85 m, z: 7.30 m) were formed in the single-phase model, while three (zone 1, x: 1.40 m z: 13.0 m; zone 2, x: 2.85 m, z: 7.30 m; zone 3, x: 4.35 m, z: 1.60 m) centers of recirculation were observed in the two-phase model, as illustrated in Figure 3 (points one, two and three). These results are also confirmed by the velocity magnitudes, i.e., close to $0.115\ \mathrm{m \cdot s^{-1}}$ at the ecosystem inlet and $0.088\ \mathrm{m \cdot s^{-1}}$ in adjacent zones for the two-phase CFD model. Based on the velocity profiles, the previous zones are directly influenced by the mixing generated from the fluid entry into the lagoon, which, in the absence of structures such as baffles or screens, causes this type of hydraulic failure [33]. The higher velocities predicted in the single-phase model (i.e., $0.127\ \mathrm{m \cdot s^{-1}}$) provide an explanation for the early release of tracer concentration and the absence of the third recirculation zone that was predicted with the two-phase model.

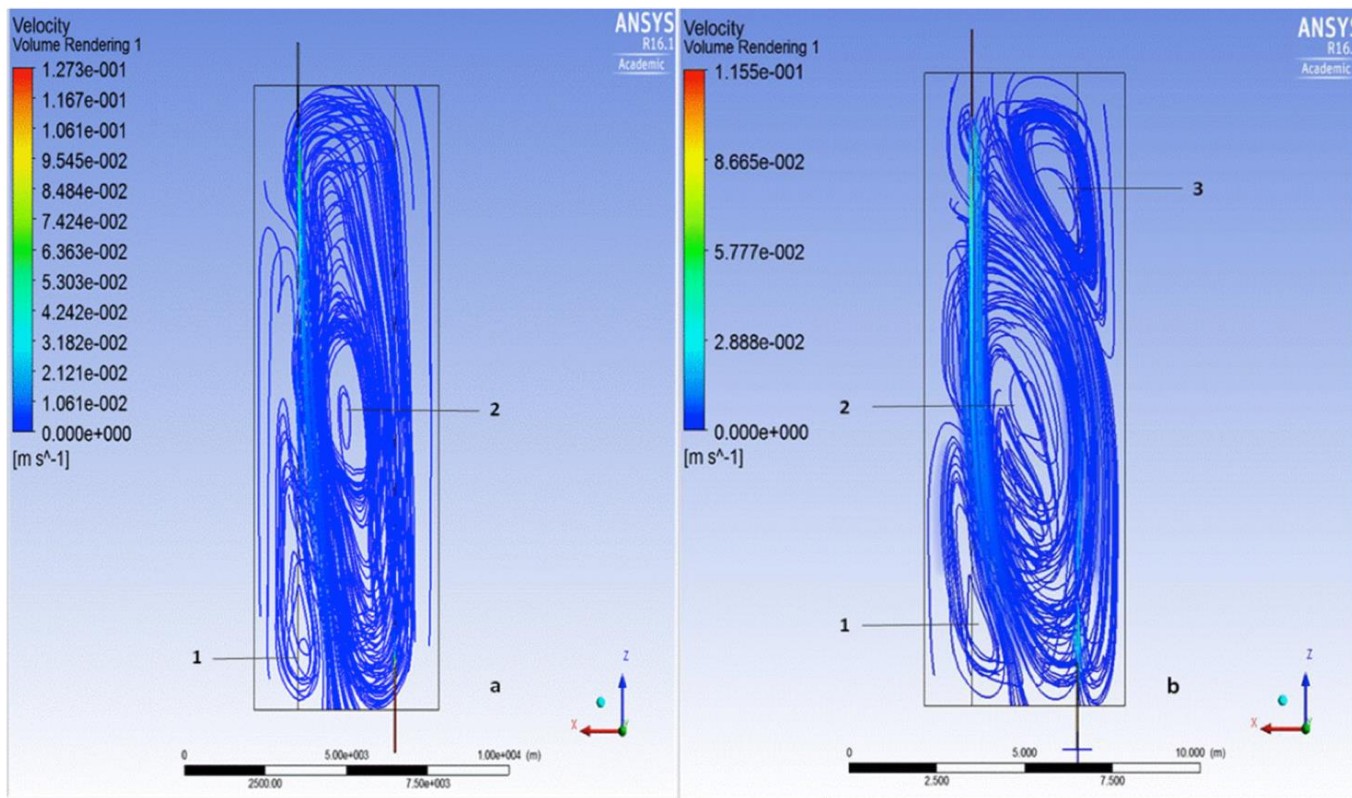

**Figure 3.** Velocity magnitudes and streamlines obtained with Fluent for the single-phase (**a**) and two-phase (**b**) models.

The simulation times and computational expenses for both models were significantly different. The convergence time for single-phase models was 50% lower than the convergence time for the two-phase model. For the latter, the simulations showed that at least 2.5 times the hydraulic retention time was required to reach stability (steady-state effluent concentration of solids and velocity profiles) and provide a closer agreement of the experimental data. The results showed that the concentration of suspended solids gradually decreased until day 5 when it reached the inlet concentration (0.041% *w/v*). Then, after day 5, the model began to simulate the actual conditions of the ecosystem. Between day 5 and day 10 of the simulation, the concentration of suspended solids in the effluent decreased until it stabilized at 0.0224% *w/v*, which corresponds to a solid elimination efficiency of 46%. This value was only 12% lower than the experimental data (52.3%). When the scalar value used for the concentration of suspended solids in the inlet FL was replaced, simulating the entry of the suspended solids through a UDF, the percentage error was reduced. The numerical results obtained with this UDF better predicted the experimental effluent suspended solids concentration, and reduced the percentage of error to 4%. It demonstrates that UDFs improve the performance of the CFD model, due to it representing the real conditions in the inlet of the facultative lagoon. Furthermore, it was demonstrated that it is necessary to characterize the influence of suspended solids concentration properly, to increase the accuracy of the effluent solids concentration. Figure 4 displays the distribution of solids inside the lagoon. As expected, the highest concentration of solids was located in the lower regions of the lagoon, and decreased as it approached the FL pond surface and near the outlet location.

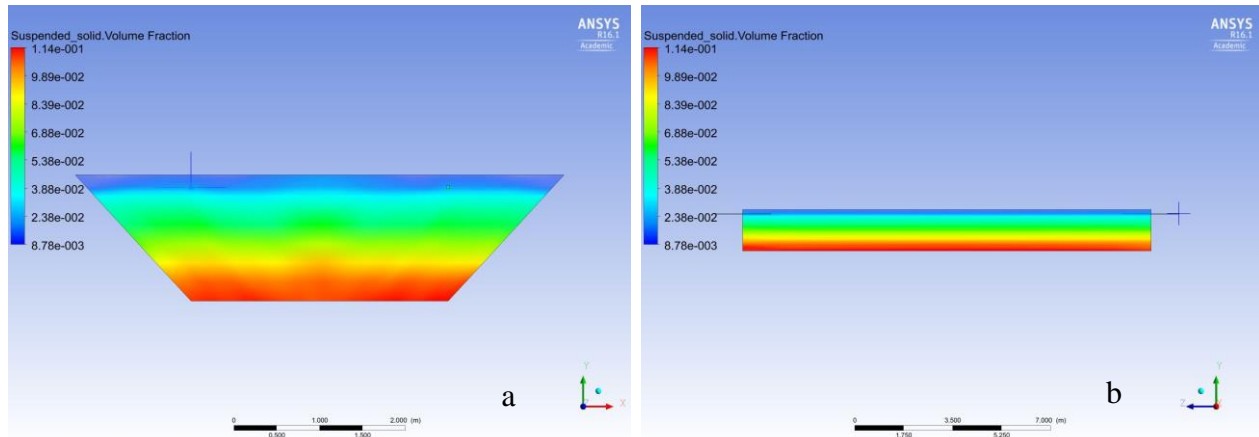

**Figure 4.** Distribution of the solids concentration in the FL: (**a**) inlet side depth wise profile (**b**) lengthwise centerline profile.

The experimental and simulated profiles of solids concentration within the FL (point L/2 and the three depths) are compared in Table 4. The best prediction of the suspended solids concentration at the different locations was achieved when the model applied the influent transient profile using the UDF. This result confirms that an accurate representation of the operating conditions of an FL is needed to predict the true separation performance expected in the lagoon.

**Table 4.** Experimental and simulated solids profile inside the facultative lagoon.

| Depth (m) | PL/2 | | | | | PL/2 | | |
|---|---|---|---|---|---|---|---|---|
| | Average Concentration | | | | | UDF "Define Profile" | | |
| | Concentration (% *w/v*) | | | | | | | |
| | Exp. | SD | Sim. | SD | %E | Sim. | SD | %E |
| 0.05 | 0.017 | 0.0023 | 0.026 | 0.0014 | 48 | 0.019 | 0.0010 | 12 |
| 0.45 | 0.016 | 0.0032 | 0.053 | 0.0037 | 230 | 0.019 | 0.0017 | 18 |
| 1.40 | 0.097 | 0.0023 | 0.094 | 0.0016 | 3 | 0.099 | 0.0009 | 2.1 |

Exp: experimental; Sim: simulated; %E: percent of error; SD: standard deviation.

The model was able to predict the lower regions of the FL, but overpredicted the upper and central regions; however, deviations lower than 20% are accepted in this type of constructed ecosystem [34] (Hernández et al. 2010). In a previous work [21], these regions were found to be the photic zone of the lagoon, where the photosynthetic activity and the interactions between the algae and microorganisms as protozoans and bacteria are presented. Therefore, the turbidimeter located in these regions measures all suspended solids, which include solids from the inlet, the algae population and microorganisms as protozoans and bacteria. In this study, the CFD model only includes the suspended solids from the inlet. This would explain the deviations at the 0.05 and 0.45 m depth. The phenomena in these regions are complex and should be taken into account in future research. The study by [35] showed that solid particles with a larger diameter (>80 μm) are deposited at the bottom of the lagoon, while those with a smaller diameter (<20 μm) are transported by the fluid to different lagoon zones (i.e., recirculation zones or until effluent outlet), which generates a gradient in the concentration of suspended solids with depth. The behavior described by [35] was observed in the concentrations of suspended solids obtained experimentally and with the two-phase model (see Table 4).

## 5. Conclusions

An evaluation of a facultative lagoon using a two-phase flow CFD model was presented. Simulation results clearly showed that capturing the mixing characteristics in these lagoons requires the modeling of solids transport. Modeling only the water flow, as has been done traditionally with single-phase models, may lead to an overprediction of the dispersion and mixing characteristics in FLs.

Numerical results in this study clearly showed that influent transient suspended solids event must be included in the model to better predict the effluent suspended solids concentration.

While the two-phase flow model was able to predict the suspended solids concentration in the lower region of the FL, the upper and central regions were overpredicted with a deviation between 12% and 18%. To improve the deviations and performance of the biphasic model in these regions, the future CFD models should include the algae population.

The validated CFD model could be used as a support tool to assess design improvements, with the intent to decrease the concentration of suspended solids within the lagoon.

**Supplementary Materials:** The following are available online at https://www.mdpi.com/article/10.3390/w13172356/s1, Table S1: Drag and lift forces models used to represent the interactions between the phases, and User Define Function (UDF) built to simulate the suspended solids concentration in the effluent of the FL.

**Author Contributions:** A.M.Z.R. CFD model development, analysis and interpretation of data and writing the manuscript; J.D. analysis and interpretation of data and review and writing the manuscript; M.R.P. CFD model development and model boundary conditions discussion; M.P. review and writing the manuscript. All authors have read and agreed to the published version of the manuscript.

**Funding:** This research received no external funding.

**Institutional Review Board Statement:** Not applicable.

**Informed Consent Statement:** Not applicable.

**Data Availability Statement:** Not applicable.

**Acknowledgments:** The authors thank Minciencias and their scholarship program, administered by Colfuturo, for the financing provided for this research project.

**Conflicts of Interest:** The authors declare no conflict of interest.

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
