# Peer review of "Computational Fluid Dynamics Simulation of Suspended Solids Transport in a Secondary Facultative Lagoon Used for Wastewater Treatment"

_water, doi:10.3390/w13172356_

Round 1
Reviewer 1 Report
Dear Authors,
Many thanks for your interesting work. I believe it is a really nice paper with a good structure. The paper has a good flow and many parts are fully connected. However, there are some issues which must be addressed in the revised version. Please find below points on your manuscript.
- The title must be changed and contracted to a more professional way!
- The novelty of the work is not fully described and not visible in your work! Only comparison of single phase and two phase is not that novel!
- You need to add a nomenclature to describe all terms and their dimensions inside that table!
- I am not quite sure that could follow your paper in term of single phase and two phase. I couldn't see that you clearly mentioned somewhere why you classify as such!
- In my opinion it is even more than two phase! How you cope with the bacteria and microorganisms! Did you consider them? Are you aware of such things as a separate phase! How you saw their effects!
- The details of the two-phase is not clear! Are you using VOF?
- Why you add something after conclusions! I suppose they refer to some supplements!
- Please provide conclusions in bullets to make them more understandable!
- The paper has a lack of fully description of the phenomenon! I think it would be better you add a schematic view of what you are talking about!
- The materials and methods part must be refined and connected and the structure must be changed! You have Experimental rig and numerical analysis! So, make a structure that the tracer and your facilities come together and the CFD part stand as another separate part!
- All of your experimental parts must be reported in details with the error sources!
- You need to have a complete validation with the possible errors between CFD and Experimental parts!
- ANSYS must be referenced properly with the date of access in your reference list as well.
- Please provide professional figures! Remove the background default blue color of ANSYS!
- Figure 1 is not a professional one! Please provide proper figure with legends!
Author Response
Please find the attached file with answers in the PDF.

Reviewer 2 Report
The paper “Characterizing the Transport of Suspended Solids in a Secondary Facultative Lagoon Using Computational Fluid Dynamics” by A. Zapata, Joel Ducoste, Miguel Peña, and Margarita Portapila addresses an interesting topic relevant to the community and to MDPI Water journal.
The authors present an evaluation of a facultative lagoon by means of a novel two-phase-flow CFD model in Ansys Fluent.
I recommend the manuscript for publication after the authors address the following questions and comments:
Major comments:
- In line 170 the authors comment about not including the algae biomass into the concentration of suspended solids and argue in the introduction that the population of algae generated in the lagoon can influence the hydrodynamics. It can be seen from the results that this assumption is valid, but do the authors think that it may be the reason of the deviation between the simulations and the experiments on the central region? Have the authors considered modelling this mass fraction in any way or would it involve a very complicated model for such a small contribution?
- In line 184 the authors state that the mixture model is used instead of the Eulerian model due to an increase of computational efficiency yet no evidence is presented to support that claim. Could the authors include some figures (computational cost vs. mesh size or whatever is more convenient) to back up their claim?
- In line 284 the authors state that “For the interactions between liquid and suspended solids, five combinations of the drag and slip forces models were tested and the results showed the same solutions for the velocity fields, number of iterations and solution convergence. Hence, the Gibilaru-Legendre-Magnaudet” combination was selected.” Could the authors include some additional information, e.g. figures, tables, etc. to support this statement?
Minor changes:
- The English of the manuscript is really good but there are some minor spelling and grammar mistakes here and there. Please do a complete revision of the manuscript and correct the errors.
- The references should be orderdered by number in order of appearance or alphabetically, the present ordering is quite confusing, so please change it accordingly.
Author Response
Please find the attachment with answers in the PDF.

Round 2
Reviewer 1 Report
Dear Authors,
Many thanks for providing the revised version. Now, as I can see the paper have been improved a lot and also many parts have been revised completely.
Some drawbacks could be the name of CFD in abstract which doesn't need any parentheses to describe. Moreover, the reference for ANSYS must be its version and the date of access! ANSYS 2013 is not acceptable! Please refer to some papers how they refer to ANSYS. AS a suggestion, It would be much better to report your contours with white or blank back ground.
Author Response
Answers to Reviewer one
Comments and Suggestions for Authors: Some drawbacks could be the name of CFD in the abstract which doesn't need any parentheses to describe. Moreover, the reference for ANSYS must be its version and the date of access! ANSYS 2013 is not acceptable! Please refer to some papers on how they refer to ANSYS. AS a suggestion, It would be much better to report your contours with a white or blank background.
Answer:
The authors are grateful to the reviewer because the comments and suggestions have been valuable to improve the quality of the document.
- The name of CFD in the abstract which doesn't need any parentheses to describe: OK it was improved
- The reference for ANSYS must be its version and the date of access! ANSYS 2013 is not acceptable! Please refer to some papers on how they refer to ANSYS: OK it was improved, all references were updated. Please see lines: 182, 221 to 227, 242, 257, 270, 301, 370, 375, 575 and 660. Please see https://www.ansys.com/academic/terms-and-conditions
- As a suggestion, It would be much better to report your contours with a white or blank background. Yes, we agree. However, due to the pandemic situation, access to the campus university in Colombia was not possible therefore get the contours with a white or blank background was no possible.
Additionally, each section of the document was improved in concordance with the suggestions and comments from the reviewer. Please see the document with “change control”
English language and style were spell-checked.

Reviewer 2 Report
In my opinion, the revised manuscript has been sufficiently improved and, thus, I recommend it for publication.
Author Response
The authors are grateful to the reviewer because the comments and suggestions have been valuable to improve the quality of the document.
English language and style were spell-checked.
